# PMIDigest: Interactive Review of Large Collections of PubMed Entries to Distill Relevant Information

**DOI:** 10.3390/genes14040942

**Published:** 2023-04-19

**Authors:** Jorge Novoa, Mónica Chagoyen, Carlos Benito, F. Javier Moreno, Florencio Pazos

**Affiliations:** 1Computational Systems Biology Group, National Centre for Biotechnology (CNB-CSIC), Darwin, 3, 28049 Madrid, Spain; 2Instituto de Gestión de la Innovación y del Conocimiento, INGENIO (CSIC and U. Politécnica de Valencia), Edificio 8E, Cam. de Vera, 46022 Valencia, Spain; 3Instituto de Investigación en Ciencias de la Alimentación (CIAL), CSIC-UAM, CEI (UAM+CSIC), Nicolás Cabrera, 9, 28049 Madrid, Spain

**Keywords:** data mining, scientific literature, literature digest, citation databases

## Abstract

Scientific knowledge is being accumulated in the biomedical literature at an unprecedented pace. The most widely used database with biomedicine-related article abstracts, PubMed, currently contains more than 36 million entries. Users performing searches in this database for a subject of interest face thousands of entries (articles) that are difficult to process manually. In this work, we present an interactive tool for automatically digesting large sets of PubMed articles: PMIDigest (PubMed IDs digester). The system allows for classification/sorting of articles according to different criteria, including the type of article and different citation-related figures. It also calculates the distribution of MeSH (medical subject headings) terms for categories of interest, providing in a picture of the themes addressed in the set. These MeSH terms are highlighted in the article abstracts in different colors depending on the category. An interactive representation of the interarticle citation network is also presented in order to easily locate article “clusters” related to particular subjects, as well as their corresponding “hub” articles. In addition to PubMed articles, the system can also process a set of Scopus or Web of Science entries. In summary, with this system, the user can have a “bird’s eye view” of a large set of articles and their main thematic tendencies and obtain additional information not evident in a plain list of abstracts.

## 1. Introduction

The contents of databases containing published biomedical literature are growing very fast. The PubMed database, which indexes abstracts for biomedicine-related articles, currently contains more than 36 million entries, and around 1.5 million new items are added each year right now (i.e., more than two papers per min), a rate that itself increases, making PubMed’s growth exponential. Consequently, there is an “information overload” in the scientific literature [1]. A particular usage of PubMed and other literature databases is to search for all articles on a particular subject in order to get an overview of that topic, for example, to start out in that research area or to write a review. Due to its large size, such a search can lead to thousands of articles, even on relatively specific subjects. For example, a simple search in PubMed for “breast cancer” results in almost half a million articles. A more restrictive search, such as “triple negative breast cancer”, produces more than 20,000 hits. Even restricting the search to review articles leads to an unmanageable number of items, as there are almost 2000 reviews on “triple negative breast cancer” (as of December 2022). Another scenario in which researchers are faced with thousands of articles to process is in the generation of systematic reviews. A systematic review is an in-depth expert critical assessment of the whole corpus of scientific literature on a given topic in order to synthesize it and extract refined conclusions, for example, reviewing the literature and clinical trials on a given disease treatment in order to generate final guidelines for practitioners (“evidence-based medicine” [2]). Systematic reviews involve not only critically evaluating large sets of articles on a given topic but also their forthcoming citation patterns in order to evaluate their impact and long-term repercussions. Consequently, the generation of systematic reviews may take a considerable amount of time and require extensive effort [3]. In these and other cases, reading and manually processing large sets of articles is difficult or even unfeasible, and tools that allow authors to automatically process, rank, digest, and extract useful information from them are increasingly demanded [4,5].

Different summarization solutions have been proposed in the literature, including the automated generation of text summaries using computational linguistic techniques [6]. Another popular approach is that based on “word cloud” representations [7,8]. This generates a graphical representation with the words (or pairs of words) within the set of articles/abstracts shown in a size proportional to their relative frequencies. In this way, words/expressions “enriched” in the set of articles (with respect to the whole bibliographic database) are visually highlighted. Another summarization strategy is to cluster the set of articles into subgroups based on automatic criteria, such as similarities of word content profiles [9], based on the idea that the resulting subgroups would have a meaning in the context of the subject represented by the original set of articles. These clustering approaches can also report features of the clusters (e.g., words differentially enriched within them) that might help in this interpretation. The screening of the initial set of articles in order to discard those not related to the subject can also be included in this category of digestion/summarization tools. These approaches are able to detect articles that, according to different criteria, do not “fit” in the set and that could be there, for example, due to artifacts of the search engine [10].

An interesting feature of PubMed is the “medical subjects headings” (MeSH) structured vocabulary: a set of keywords dealing with different biomedical aspects that are manually assigned to articles [11]. While the digestion/summarization approaches discussed above can, in principle, be applied to any set of articles/abstracts, some specific tools take advantage of the MeSH vocabulary in order to process sets of PubMed entries (e.g., [12,13,14]). These derive different statistics from the frequencies of MeSH terms and link articles based on the co-occurrence of term contents. They also provide advanced visualization capabilities to inspect these MeSH contents.

Representing large datasets as complex networks of nodes connected by relationships is a strategy used in many disciplines to visualize these datasets and extract information from the network topological parameters [15]. This kind of network representation is widely used in molecular biology and biomedicine to represent large sets of relationships between bioentities [16]. Network approaches have also been used to visualize and analyze literature datasets. In these representations, the nodes are usually the articles, and the links represent different types of relationships between them, such as common keywords (e.g., genes, MeSH terms, etc.) [17], similar word contents, or citations from one article to the other [18]. In these literature-derived networks, nodes can also represent other entities linked by the information contained in the set of articles, such as gene–gene relationships obtained by comentions in the literature [19].

In this work, we present PMIDigest, a general-purpose tool for distilling large biomedicine article datasets and presenting the summarized information as an interactive graphical web report. In PMIDigest, we implemented features that were useful in our own work generating systematic reviews.

## 2. Materials and Methods

The simplest input for PMIDigest is a text file with the list of PubMed identifiers (PMIDs) for the set of articles of interest. As examples, this set of articles can be the result of a search on PubMed’s web interface or of automatic calls to the NCBI Entrez API (Figure 1). In the case of an interactive search on PubMed’s web interface, the file with the plain list of PMIDs required by PMIDigest can be obtained with the “Save” option, choosing “All Results” in “Selection” and “PMID” in “Format”. The PMIDigest distribution (see below) includes examples of this input file.

Files with sets of articles from other bibliographic resources, such as Scopus or Web of Science (WoS), can also be imported if they are provided in a specific XML format described in the documentation. These can be combined with sets of PMIDs in order to inspect sets of articles from different sources together. In the case of articles from Scopus, WoS, or other resources, some PubMed-specific data shown in the interface will not be available, such as the MeSH keywords or citation data. Examples of these files are also included in the distribution.

PMIDigest automatically downloads the required data for the PubMed articles through the NCBI Entrez API and stores them locally so as to avoid retrieving data on the same article(s) if these are required later. These data include the set of citing articles in PubMed Central (PMC) from which different citation figures are calculated. For other bibliographic resources, these data have to be provided in the XML files themselves (Figure 1).

As additional input, PMIDigest processes the whole MeSH vocabulary, searching for the terms of certain categories of interest for the user (e.g., microorganisms, diseases, etc.) (Figure 1). These categories are specified as input by the user in the form of MeSH “semantic types”. For MeSH terms related to microorganisms, links to the NCBI Taxonomy database [20] are generated, while those related to chemical compounds are linked to the corresponding entries in the CAS registry and the FDA Substance Registration System.

The data downloaded for the PubMed entries includes their PMID, title, abstract, reference, date, authors, article type, citing articles, MeSH terms, and links to clinical trial databases if available.

Pattern-matching approaches are used to locate mentions of MeSH terms (as well as generic user-defined terms) in the articles’ abstracts and titles. This includes the terms’ synonyms defined in the resource.

The retrieved citation information is used to generate a “citation network”. In this network, nodes are articles, and directed links connect citing articles with cited articles.

After processing all this information, an interactive web-based report is generated as a local HTML file that can be visualized offline in any web browser. This interactive HTML report uses open-source JavaScript libraries described in the documentation, including Cytoscape’s JS libraries for network visualization [21].

PMIDigest is distributed as a command line tool coded in the Perl programming language, together with other files required by the interactive web page. It works on Linux, Windows, and MacOS. The free Perl interpreter has to be installed. The interactive HTML report generated by the program can be visualized in the main web browsers: Firefox, Chrome, Safari, and MS Edge. The open-source package wget for the retrieval of content from web servers is also required. This package is freely available for all operating systems mentioned above. The distribution includes installation instructions, documentation, examples, and a help file accessible from the same interface. PMIDigest can be freely downloaded from https://github.com/JNovoaR/PMIDigest, accessed on 13 March 2023. An example interactive report is available online at https://csbg.cnb.csic.es/jnovoa/PMIDigest/example/FoodAdditives.html, accessed on 13 March 2023.

## 3. Results

### 3.1. Interface Sections

Screenshots of the interactive web report generated by PMIDigest are shown in Figure 2. 

#### 3.1.1. Article List

The top-left panel contains a table with the list of articles within the input dataset. The first column shows the reference and the PubMed identifier (PMID), which is a link to the PubMed record for that entry. The second column shows the source of that item: PubMed (PM), Web of Science (WoS), or Scopus. The next column indicates the type of article, highlighting three types of special interest in biomedicine: reviews (R), systematic reviews (S), and clinical trials (C). In the following columns, the publication date and the title of the article are shown. The next three columns contain citation-related figures: the total number of PMC citing articles, the average number of citations per year, and the number of citations from articles within the same dataset. The citations/year parameter provides an idea of how trendy an article is and can point out to promising articles published recently (therefore without a large number of total citations) but that are highly cited. Similarly, the number of “internal” citations (from other articles within the same dataset) can help to identify important articles for the particular subject the dataset is representing. The two last columns show the tags assigned to the articles and a check box for selecting and manipulating them (see below). When an article (row) is selected in this list, the corresponding node becomes highlighted in the “Citation network” panel, and its details are shown in the “Details” panel (see below). The table can be sorted by any column by clicking the corresponding header. This allows, for example, for inspection of the most recent articles (sorting by date) or those with the most citations or for compilation of all articles of a given type (e.g., reviews) or with a given tag (see below).

#### 3.1.2. MeSH Terms List

The bottom-left panel shows the list of MeSH terms from the categories of interest for the user assigned to the articles of the dataset, together with their frequencies within the dataset (number of articles linked to each term). The list is sorted by those frequencies, and the descriptions of the MeSH terms are shown in different colors depending on the categories they belong to, using a color scheme defined by the user. The MeSH identifiers are links to the corresponding entries in the MeSH resource. For MeSH terms related to microorganisms, a link to the NCBI Taxonomy database [20] is included, and those related to chemical compounds are linked to different chemical-related resources. The last column contains a link to expand a list with the entries (PMIDs) associated with a particular MeSH term, which are links to show the corresponding article in the “Citation network” and “Details” panels (see below). Among other things, this panel allows the user to obtain an overview of the main thematic tendencies of the set of articles—usually represented by the MeSH terms with the highest frequencies. Apart from the keywords defined in the MeSH vocabulary, this list can also show a generic set of terms specified by the user. These user-defined terms are shown in purple in this list (see “Operations with Articles” below).

#### 3.1.3. Citation Network

The bottom-right panel shows the citation network for the articles in the input set. Each node represents an article, and the (directed) links represent citations from one article to another. Nodes are colored according to the number of internal citations received, from dark blue (fewer) to yellow (more). Papers that do not participate in any internal citation (either citing or being cited) are not included in this network. The representation can be zoomed-in/out with the mouse wheel and moved by dragging the mouse in the background. It is also possible to move the nodes by dragging them with the mouse. Right clicking a node highlights its incoming and outgoing links (citations) in blue and red, respectively (Figure 2). Clicking a node with the left button shows the details of the corresponding paper in the “Details” panel (see below). As mentioned above, nodes are also highlighted when the corresponding articles are selected in the “Article list” or “MeSH” panels. In these cases, the display automatically moves to those nodes if they are out of view, due to, for example, a large zoom.

A network layout is the placement of network nodes according to different criteria for representation purposes only. Different layouts can be used to represent the same network, depending on which feature the user wants to highlight. In this panel, several graph layouts can be chosen with the “[+]” menu. 

This network allows, for example, for detection of “clusters” (groups of papers citing each other recurrently while not citing others in the dataset). These “citation clusters” might represent different subthemes within the general subject represented by the set of papers, different points of view on the topic, etc. The node color allows for detection of the “hubs” of the clusters (i.e., the central, highly cited papers), which could be good starting points to dig into these subthemes or points of view represented by the clusters. 

The citation network is not available for collections of more than 3000 articles due to its computational cost. 

#### 3.1.4. Article Details

Finally, the top-right panel shows the details of an article selected in any of the other panels (list, MeSH, or network). This section displays the title and abstract of the selected article. MeSH terms from the user-defined categories and generic terms introduced by the user are highlighted and colored as in the MeSH panel. Although MeSH terms are only associated with PubMed articles, their mentions in articles from other sources (Scopus and WoS) are also highlighted. Similarly, for a particular PubMed entry, all mentions of any MeSH terms in its title/abstract within the whole set are highlighted, irrespective of whether they are indexed for that particular article or not. This section also includes the reference and type of article at the top, as well as the article PMID, which is a link to the article record in the corresponding database. If an article is associated with entries in clinical trial databases, links to the corresponding records in these databases are shown at the bottom of the panel.

### 3.2. Operations with Articles

The interface incorporates some features for ranking and filtering the set of articles.

It is possible to assign user-defined tags (labels) to the articles, which are shown in the “Tag” column of the article list. To tag papers, select one or more items from the article list by checking the corresponding checkboxes; then, either tag these articles as “important” (“Tag as Imp.” button) or use custom tags. Rows for articles tagged as “important” are highlighted with a green background. The “Other tags” button allows the user to define new generic tags and assign them to the selected articles. It also allows for previously defined tags to be assigned the selected articles. The “Untag” button removes the tags from the selected papers.

The “Del. selection” button allows the selected papers to be (temporarily) removed from the article list. These go to a “Trash” list accessible from the corresponding button, from where they can be eventually selected and recovered with “Undelete selection” button.

Apart from the MeSH terms mentioned earlier, it is possible for the user to define a set of generic terms of interest with the “Enter new terms” button. These terms are highlighted in purple and added to the panel along with the list of MeSH terms, together with their frequencies in the set of articles.

### 3.3. Other Features

Using the browser’s “Save” function, it is possible to save the report in HTML format, including all the changes made (deleted articles, new tags and terms introduced, table sorting, etc.) so that the working session can be resumed later by opening that HTML file.

The interface includes a link to the local help file included in the distribution, which describes all functionalities.

## 4. Discussion

The pace at which bibliographic databases are growing pushes the development of automatic tools for processing such massive amounts of information. The amount of bibliographic information on a given subject, even if very specific, is very large and therefore difficult to process manually. Although the landscape of tools for this general purpose is large and continuously changing as new tools emerge, we can classify them into two main families: tools for the targeted extraction of specific types of information and summarization approaches aimed at helping in the processing of large text corpora but without an particular intended type of information in mind.

There are many automatic tools for the “targeted” extraction of information from the biomedical scientific literature. These tools process raw text and, using text-mining approaches, detect mentions of particular entities of biological interest (e.g., genes, proteins, diseases, etc.) [22], as well as action verbs and other context information, and report them in a structured manner (e.g., “protein_A interacts_with protein_B” [23]). Some of these approaches use advanced “natural language processing” (NLP) methods to detect the roles of words in sentences in order to facilitate the detection of specific patterns indicative of the specific information they are looking for [24]. These text-mining approaches can also be used to extract other very specific data of interest for particular projects, such as the number of patients/controls in a clinical trial report [4]. Many of these tools define an a priori “ontology” (representation of knowledge in a computer-tractable way) and scan the literature based on it (e.g., [25]). For these reasons, these tools are tailored to specific scenarios, as they are targeted at the retrieval of specific pieces of information and are usually domain-oriented (intended for a specific area/discipline).

An alternative (and complementary) approach is to use summarization techniques that help to digest generic literature corpora without looking for a particular type of information. However, there are not many open-source tools available to summarize large generic sets of articles. The most powerful approaches are commercial solutions, such as DistillerSR, whereas the landscape of open-source tools is still limited and focused on specific problems. The tool described in this work falls in this category.

The application of machine learning approaches is revolutionizing scientific research in general and biomedicine in particular [26], with the recently developed system AlphaFold for the prediction of high-quality 3D structural models for proteins being the most prominent example [27]. Very recently, a family of machine learning approaches called “generative language models” (e.g., ChatGPT) has become very popular due to their astonishing capacity to learn language structures from written text and use them to generate answers to questions that read like human-written text. The potential of this approach for text mining and text summarization is very high, and we are see applications in these areas in the future. Meanwhile, tools such as that presented here can help users with these issues.

## 5. Conclusions

We developed a generic tool for digesting large sets of biomedicine-related articles that generates a user-friendly, interactive, and dynamic web report with features intended for the user to obtain a “bird’s eye view” of the main topics and subtopics of the set of articles, their overall citation tendencies, internal citation relationships, etc. Wherever possible, links to the original article and to external resources with additional information are provided.

In the toll described herein, we implemented features we found useful in our own work generating systematic reviews. For that reason, we think PMIDigest can be useful in different scenarios, such as the generation of systematic reviews or for initiation into a specific subject via related bibliographic searches.

In the future, we plan to add new features to the tool, such as the possibility of processing sets of clinical trial records, the generation of printable reports for a generic set of articles, the generation of word clouds, or the automatic analysis of the citation network.

## Figures and Tables

**Figure 1 genes-14-00942-f001:**
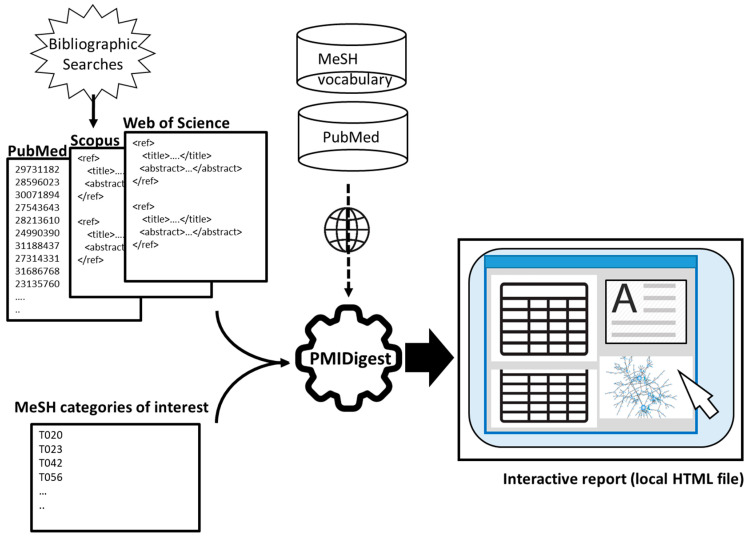
PMIDigest workflow for generating an interactive web report. The user input is shown on the left: sets of articles from PubMed, WoS, Scopus, or other resources (for example, from bibliographic searches), together with the MeSH categories of interest. This input, together with information retrieved automatically from PubMed and MeSH, is used to generate the interactive web report as a local HTML file.

**Figure 2 genes-14-00942-f002:**
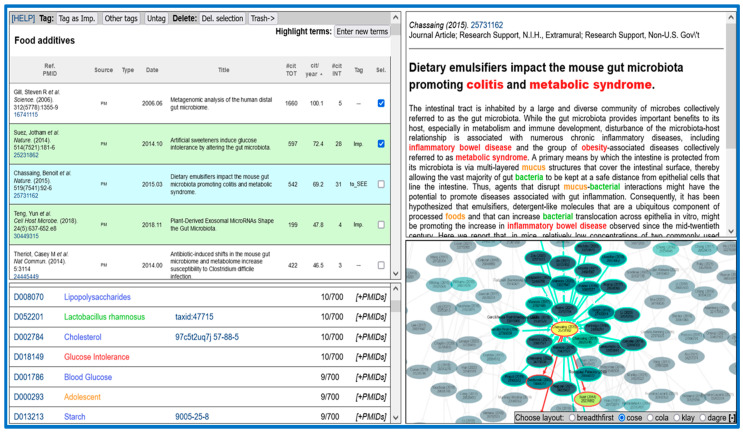
Screenshot of a PMIDigest interactive web report. The four main panels are: a table with the list of articles (**top left**), a table with MeSH terms (**bottom left**), the citation network (**bottom right**), and article details (**top right**). In this example, MeSH terms related to diseases are colored red; those describing microorganisms are indicated in green; those related to chemical and biological compounds are indicated in blue; and those associated with food, body parts/substances, and age are indicated in orange. The selected article is highlighted in the article list (light-blue background) and the citation network (with the incoming citations shown as light-blue arrows and the outgoing citations as red arrows), and its details (including the abstract) are shown in the top-right panel. The interactive report for this example is available at https://csbg.cnb.csic.es/jnovoa/PMIDigest/example/FoodAdditives.html, accessed on 13 March 2023.

## Data Availability

The software described in this article, including documentation and examples, is freely available at https://github.com/JNovoaR/PMIDigest, accessed on 13 March 2023. An interactive report with an example generated by the program is available at https://csbg.cnb.csic.es/jnovoa/PMIDigest/example/FoodAdditives.html, accessed on 13 March 2023.

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
