# Peer review of "PMIDigest: Interactive Review of Large Collections of PubMed Entries to Distill Relevant Information"

_genes, 2023, doi:10.3390/genes14040942_

Round 1

Reviewer 1 Report

Jorge Novoa and colleagues have developed a novel tool for accessing, organizing, and presenting large sets of biomedicine-related articles from a vast pool of research articles available on PubMed dattabase based on generic or user specific scientific terms. This tool has an impressive interface which links not only the article source such as, PubMed, Web of Science or Scopus but also the NCBI taxonomy database as well as different chemical-related resource. The selling point of this tool is that it’s user-friendly, interactive and provides a holistic report for the user to obtain a detailed overview of the main topics and subtopics of the set of articles, their overall citation tendencies as well as internal citation relationships, etc. Overall, this paper is well written and the description of each of module and key provided in this web tool are well explained with figures. I recommend a few minor corrections that could improve the overall paper quality.

Comment 1:  The sentence in line 92, “systematic revies" the word revies is to be corrected to reviews.

Comment 2: The sentence in line 113-114, “These categories are specified as input by the user in the form of MeSH “semantic trypes” the word trypes is to be corrected to “types”.

Comment 3: In line 181, “the descriptions of the MeSH terms are colored according with the categories they belong to” needs to be rephrased.

Comment 4: In line 187, “this panel allows to easily grasp the main thematic tendencies of the set of articles” needs to be rephrased.

Comment 5: In line 188-190, “Apart from the keywords defined in the MeSH vocabulary, this list can also show a generic set of terms specified by the user (see below).” Which generic terms are the authors referring to in the bottom left panel of Figure 2 and what is being shown in (see below) please elaborate on it.

Comment 6: In line 201, “(placement of the nodes according with different criteria)” needs to be rephrased.

Author Response

Jorge Novoa and colleagues have developed a novel tool for accessing, organizing, and presenting large sets of biomedicine-related articles from a vast pool of research articles available on PubMed dattabase based on generic or user specific scientific terms. This tool has an impressive interface which links not only the article source such as, PubMed, Web of Science or Scopus but also the NCBI taxonomy database as well as different chemical-related resource. The selling point of this tool is that it’s user-friendly, interactive and provides a holistic report for the user to obtain a detailed overview of the main topics and subtopics of the set of articles, their overall citation tendencies as well as internal citation relationships, etc. Overall, this paper is well written and the description of each of module and key provided in this web tool are well explained with figures. I recommend a few minor corrections that could improve the overall paper quality.

We thank the Referee for these encouraging comments.

Comment 1:  The sentence in line 92, “systematic revies" the word revies is to be corrected to reviews.

Comment 2: The sentence in line 113-114, “These categories are specified as input by the user in the form of MeSH “semantic trypes” the word trypes is to be corrected to “types”.

Comment 3: In line 181, “the descriptions of the MeSH terms are colored according with the categories they belong to” needs to be rephrased.

Comment 4: In line 187, “this panel allows to easily grasp the main thematic tendencies of the set of articles” needs to be rephrased.

All these spelling mistakes above were corrected. Thanks for pointing them out. All the required re-phrasings were also introduced.

Comment 5: In line 188-190, “Apart from the keywords defined in the MeSH vocabulary, this list can also show a generic set of terms specified by the user (see below).” Which generic terms are the authors referring to in the bottom left panel of Figure 2 and what is being shown in (see below) please elaborate on it.

Apart from the MeSH terms, a generic set of terms specified by the user can also be added to that list and highlighted in the articles. Now, for the revised version, in that sentence we refer the reader to the manuscript section where these “user-defined” terms are explained in detail.

Comment 6: In line 201, “(placement of the nodes according with different criteria)” needs to be rephrased.

We expanded the description of “network layouts” in that part of the manuscript. We hope it is clearer now.

Reviewer 2 Report

The paper entitled "PMIDigest: Interactive review of large collections of PubMed 2 entries to distill relevant information” studies interactive tool for automatically digesting big sets of PubMed articles.

I found the article to be interesting and insightful. I think the authors have tried to put their findings into context. The major points are the following:

1. The current title is misleading, and unclear (PMIDigest).  I recommend to explain it.

2. The contributions are not mentioned clearly. “In this work we present PMIDigest, a general-purpose tool for …… systematic revies” (Its must be “systematic reviews”). I suggest to add the main contributions as a list. It would be easy for the reader of this journal to caught your contribution easily.

3. Add the literature sections, mostly in reviews this section contained the important information, use paragraph with main ideas instead.

4. The author explained the Figure 1 well. However, I recommend to provide high-quality Figure. If the author only provide the Flowchart of this Figure, and also give explanation. Additionally, the interactive report (top-left, and bottom-left) in Figure 1 must be provided in Table form rather than screenshot, that would be more understable. To represent the “Vitamins D ….. Colitis” (top-right, and bottom-right) in interactive report use standard apps or python to represent the statistics.

5. The term “The information retrieved” is provide in Page 4 line 125. The Information retrieval is separate research interest area. Its is recommended to explain it in introduction or literature Section. It is just abrupt in this section which is unclear for the journal readers. Provide the references that’s related to your title and most recent.

6. Figure 2 has the same issue like Figure 1. Follow the same instruction for Figure 2. Provide (table with the 167 list of articles (top-left), table with MeSH terms (bottom-left) in separate Table, while citation network (bottom-right) and 168 article details (top-right) in Figure form).  

7. I admit that the investigation of using biomedical large collections of PubMed 2 entries to distill relevant information is interesting and unexplored. However, I didn't see detailed and comprehensive discussions or analyses of this topic although the authors claim to tackle it in their title.

8. It is recommended to provide the Challenges and Future work as a separates Section. Additionally, In review paper most recent works is represented, with solid proof.

9. The conclusion part is missing.

1. No tables about biomedical (PubMed) data. Table of general-domain datasets is presented only, but no statistic or sample is provided.

Author Response

The paper entitled "PMIDigest: Interactive review of large collections of PubMed 2 entries to distill relevant information” studies interactive tool for automatically digesting big sets of PubMed articles.

I found the article to be interesting and insightful. I think the authors have tried to put their findings into context. The major points are the following:

Thanks for the comments.

  1. The current title is misleading, and unclear (PMIDigest). I recommend to explain it.

PMIDigest = PubMed Identifier (PMID) + “digester”. We added that to the revised Abstract.

  1. The contributions are not mentioned clearly. “In this work we present PMIDigest, a general-purpose tool for …… systematic revies” (Its must be “systematic reviews”). I suggest to add the main contributions as a list. It would be easy for the reader of this journal to caught your contribution easily.

“Review” corrected, thanks. That part of the Introduction tries to summarize the tool’s objectives in a single sentence. Its features (is that what the Referee means by “contributions”?) are listed in the Abstract and detailed through the manuscript.

  1. Add the literature sections, mostly in reviews this section contained the important information, use paragraph with main ideas instead.

We are not sure of understanding what the Referee means here. Do you mean displaying the PubMed articles’ cited references in detail? That information is not available for all PubMed entries. We focused on fields that are (in principle) available for all entries.

  1. The author explained the Figure 1 well. However, I recommend to provide high-quality Figure. If the author only provide the Flowchart of this Figure, and also give explanation. Additionally, the interactive report (top-left, and bottom-left) in Figure 1 must be provided in Table form rather than screenshot, that would be more understable. To represent the “Vitamins D ….. Colitis” (top-right, and bottom-right) in interactive report use standard apps or python to represent the statistics.

Following yours as well as other Referees’ comments we changed that figure to a more schematic representation and removed the screenshot for clarity. A more detailed screenshot of the interactive interface is available in Figure 2, and the table with all the information requested by the referee is available online (see answers to your comments on Figure 2 below).

We are not sure of understanding the last sentence of this comment.

  1. The term “The information retrieved” is provide in Page 4 line 125. The Information retrieval is separate research interest area. Its is recommended to explain it in introduction or literature Section. It is just abrupt in this section which is unclear for the journal readers. Provide the references that’s related to your title and most recent.

The Referee is right that “information retrieval” is a standard term for a research area with specific goals and methodologies. We changed that to “data download” for the revised version.

  1. Figure 2 has the same issue like Figure 1. Follow the same instruction for Figure 2. Provide (table with the 167 list of articles (top-left), table with MeSH terms (bottom-left) in separate Table, while citation network (bottom-right) and 168 article details (top-right) in Figure form).

These data requested by the Referee are available at the interactive interface with the example from where that screenshot was taken, whose URL is provided in the figure legend (https://csbg.cnb.csic.es/jnovoa/PMIDigest/example/FoodAdditives.html). There you can interactively browse the full article list, list of MeSH terms, etc. In any case, this figure is for illustrative purposes only. The reader does not need to see the whole list of articles/MesH terms in order to understand it.

  1. I admit that the investigation of using biomedical large collections of PubMed 2 entries to distill relevant information is interesting and unexplored. However, I didn't see detailed and comprehensive discussions or analyses of this topic although the authors claim to tackle it in their title.

The features of the interface were designed with that goal in mind: help the user distilling relevant information from a large collection of biomedical articles, as an alternative to read all articles in detail, what could be very time consuming or even unfeasible. We changed some parts of the Introduction and Discussion trying to make this clearer.

  1. It is recommended to provide the Challenges and Future work as a separates Section. Additionally, In review paper most recent works is represented, with solid proof.

  1. The conclusion part is missing.

For the revised version, we integrated the conclusions and discussion into a single section and renamed it to “Discussion and Conclusions”. There we also empathize the challenges and intended future work.

  1. No tables about biomedical (PubMed) data. Table of general-domain datasets is presented only, but no statistic or sample is provided.

We do not understand this point. Sorry about that. As explained above, an example interactive report is available online to allow the reader getting a better grasp of all functionalities and information provided by our tool without the need of installing it (https://csbg.cnb.csic.es/jnovoa/PMIDigest/example/FoodAdditives.html). We do not know if that covers the Referee’s comment.

Reviewer 3 Report

The manuscript "PMIDigest: Interactive review of large collections of PubMed entries to distill relevant information" is quite interesting and perfectly written. The methodology, results and conclusion sections are solid.

Author Response

The manuscript "PMIDigest: Interactive review of large collections of PubMed entries to distill relevant information" is quite interesting and perfectly written. The methodology, results and conclusion sections are solid.

We thank the Reviewer for the positive comments.

Reviewer 4 Report

Novoa et al. developed PMIDigest, which has the potential to be useful in different ways, like helping to create systematic reviews or allowing people to explore a particular subject by searching for relevant bibliographic information. I have a few concerns.

(1) What is the novelty of this method? There are similar tools/approaches available, such as Textpresso (https://doi.org/10.1371/journal.pbio.0020309), BioRAT (https://doi.org/10.1093/bioinformatics/bth386), PubMedMiner (https://www.ncbi.nlm.nih.gov/pmc/articles/PMC4419975/), etc. The authors should address these tools in their paper and explain how PMIDigest differs from these tools and is better.

(2) The author should add a conclusion section, which should highlight the potential application and future scope.

(3) The workflow (Figure 1) should be better and more informative.

(4) The discussion section needs to be revised as it is not well-written.

(5) The authors did not provide a thorough explanation of how the text-mining approach works in their method. They need to give a detailed description in the Materials and Methods section.

Author Response

Novoa et al. developed PMIDigest, which has the potential to be useful in different ways, like helping to create systematic reviews or allowing people to explore a particular subject by searching for relevant bibliographic information. I have a few concerns.

Thanks for the comment.

(1) What is the novelty of this method? There are similar tools/approaches available, such as Textpresso (https://doi.org/10.1371/journal.pbio.0020309), BioRAT (https://doi.org/10.1093/bioinformatics/bth386), PubMedMiner (https://www.ncbi.nlm.nih.gov/pmc/articles/PMC4419975/), etc. The authors should address these tools in their paper and explain how PMIDigest differs from these tools and is better.

We already cited PubMedminer [12]. We were aware of the other two but they are text-mining approaches aimed at detecting specific pieces of information (i.e. look for a particular/specific type of information), and not summarization approaches aimed at facilitating the generic digestion of literature sets, as the one presented here. We expanded the discussion on these two complementary approaches in the new “Discussion/Conclusions” section and hence we have introduced these and other new references there.

(2) The author should add a conclusion section, which should highlight the potential application and future scope.

For the revised version, we have restructured the Discussion section into “Discussion and Conclusions”, and highlighted these issues on potential applications and future scope. We hope it is clearer now.

(3) The workflow (Figure 1) should be better and more informative.

We have changed the Figure 1 to make it more schematic and removed the real screenshot. We also added some other elements to make the workflow clearer.

(4) The discussion section needs to be revised as it is not well-written.

We have fully revised that section and integrate it with “Conclusions”. We hope it’s clearer now.

(5) The authors did not provide a thorough explanation of how the text-mining approach works in their method. They need to give a detailed description in the Materials and Methods section.

There is no “sophisticated” text-mining in our approach. We do not classify our tool as such. We only mention text-mining in the Discussion/Conclusions when commenting other approaches/tools for processing large text corpuses. As commented in the answers above, this section was rewritten, among other things, to make this distinction clearer. In any case, prompted by the Referee’s comment, we include some more details on the Methods section.

Round 2

Reviewer 2 Report

The paper entitled " PMIDigest: Interactive review of large collections of PubMed 2
entries to distill relevant information” studies proposed classifying or sorting the articles.
I found the article interesting, insightful, and particularly well-revised. I think the authors have tried to put their findings into context. 

Specific Comments:

Abstract: No comments; well-written.

 Introduction:  No comments; well-written.

Material and Method: No comments; well-written.

Results: No comments; well-written.

Conclusions: I recommend to separates the conclusion from the discussion. Add the separate section "Conclusion and future work".

Reviewer 4 Report

While the authors made significant improvements to the manuscript, I would not suggest combining the Discussion and Conclusion sections. It would be better to keep these two sections separate.